# Decentralized study of COVID Vaccine Antibody Response (STOPCoV): Results of a participant satisfaction survey

Rizani Ravindran[1], Leah Szadkowski[2], Leif Erik Lovblom[2], Rosemarie Clarke[1], Qian Wen Huang[1], Dorin Manase[3], Laura Parente[4], Sharon Walmsley[1,5,6]*, on behalf of the STOPCoV research team[¶]

1 University Health Network, Toronto General Hospital Research Institute, Toronto, Ontario, Canada, 2 Biostatistics Research Unit; University Health Network; University of Toronto, Toronto, Ontario, Canada, 3 DATA Team; University Health Network; University of Toronto, Toronto, Ontario, Canada, 4 Health care Human Factors; University Health Network; University of Toronto, Toronto, Ontario, Canada, 5 Faculty of Medicine, University of Toronto, Toronto, Ontario, Canada, 6 Department of Medicine, University Health Network; University of Toronto, Toronto, Ontario, Canada

¶ Membership of the STOPCoV research team is provided in the acknowledgement
* Sharon.walmsley@uhn.ca

**Data Availability Statement:** One of the study funder, the COVID-19 Immunity Task Force (CITF), has a data sharing protocol for all funded projects.

## Abstract

The Covid-19 pandemic required many clinical trials to adopt a decentralized framework to continue research activities during lock down restrictions. The STOPCoV study was designed to assess the safety and efficacy of Covid-19 vaccines in those aged 70 and above compared to those aged 30–50 years of age. In this sub-study we aimed to determine participant satisfaction for the decentralized processes, accessing the study website and collecting and submitting study specimens. The satisfaction survey was based on a Likert scale developed by a team of three investigators. Overall, there were 42 questions for respondents to answer. The invitation to participate with a link to the survey was emailed to 1253 active participants near the mid-way point of the main STOPCoV trial (April 2022). The results were collated and answers were compared between the two age cohorts. Overall, 70% (83% older, 54% younger cohort, no difference by sex) responded to the survey. The overall feedback was positive with over 90% of respondents answering that the website was easy to use. Despite the age gap, both the older cohort and younger cohort reported ease of performing study activities through a personal electronic device. Only 30% of the participants had previously participated in a clinical trial, however over 90% agreed that they would be willing to participate in future clinical research. Some difficulties were noted in refreshing the browser whenever updates to the website were made. The feedback attained will be used to improve current processes and procedures of the STOPCoV trial as well as share learning experiences to inform future fully decentralized research studies.

We will transfer relevant anonymized study data as available to the CITF as a part of these standard data sharing requirements. This is submitted together with a data dictionary defining each field in the set. External researchers will be able to submit a request to the CITF to receive access to all CITF data through their data access committee. The CITF will employ a rigorous checklist to ensure that these external requests follow all necessary ethical and privacy protocols. The data provided to the CITF will be stored on the CITF Database. The data on the CITF Database will be held under the custodianship of McGill University or one of its collaborators and be shared via the cloud, both nationally and internationally. Data in the CITF Database can be used by researchers across Canada and in other countries following Data Access Committee (DAC) approval. These transfers will also be made in compliance with Canadian law and research ethics. A DAC will be responsible for reviewing applications for access to the data and for approving applications that respect the privacy and access policies of the CITF. The DAC will require that researchers confirm that their intended research activities have received necessary ethics approvals. The data may also be shared with other COVID- 19 research databases that follow similar protections and procedures as the CITF Database. Further the main study protocol, statistical analysis plan, informed consent form and full protocol are available on the study website www.stopcov.ca.

**Funding:** This study was funded in part by the Public Health Agency of Canada through a grant from the Canadian Immunity Task force (CITF ID: 087-VS), the Canadian Institutes of Health Research (EG2-179431), and through a contribution from the Speck family through the University Health Network Foundation (Speck Family COVID-19 Research) to SW as the principal investigator. The funders had no role in study design, data collection and analysis, decision to publish or preparation of the manuscript.

**Competing interests:** I have read the journal's policy and the authors have the following competing interest: SW has served on advisory boards, spoken at CME events and received research funds from ViiV, GSK, Merck, and Gilead pharmaceutical companies.

## Author summary

We report on the results of a satisfaction survey completed by participants of the STOP-CoV study. The former is an ongoing 96 weeks study of the safety and antibody responses to COVID vaccines in Ontario Canada. This was a completely decentralized study in which participants used a unique study identification number and password to connect to a digital platform wherein they completed their consent, questionnaires, obtained information about the study procedures, viewed videos, recorded vaccine doses, symptoms and dates of sample collection and receive results. It enrolled participants > 70 years and compared to those 30–50 years. In this sub-study participants completed a satisfaction survey about the digital platform, ease of use, understanding of procedures and questionnaires. Overall, despite a minority previously participating in research the satisfaction with the study was high, and generally the procedures were felt to be understandable and easy to use. Having contact with the study team by email or telephone was important to answer questions around digital issues such as refreshing the browser for study updates. The use of the platform enabled a large number of elderly participants to remain engaged and contribute from their homes, often remote from academic study centers.

## Introduction

Beginning in January 2020, the COVID-19 pandemic caused unprecedented illness and death, economic crisis and severe disruption to normal daily life globally and in Canada [1,2]. The elderly and those with comorbidity were amongst the most vulnerable [3]. Preventative vaccines were made available at exceptional speed and the initial protective rate of 90–94% for the 2-dose messenger RNA vaccines was far better than could have been anticipated [4,5]. The clinical trials, although adequately powered and of gender parity, primarily enrolled white persons < 70 years of age with minimal comorbidity not reflective of the Canadian population. Questions remained including the protective antibody level, the extent and durability of the vaccine serologic response, long term efficacy, the risk of breakthrough infection and the impact on any emerging new variant strains [6–14]. Further, the National Advisory Committee on Immunization in Canada recommendations on longer delays between vaccine doses [15] than used in the clinical trials in attempt to partially immunize more of the population added further uncertainty [9,16–19].

Faced with these questions, we felt an urgent unmet need to determine the long term antibody response data to COVID-19 vaccines outside of the clinical trial setting to inform the potential need for boosters particularly among the elderly. Similar work was being conducted in long term care facilities where the death rate from COVID-19 infection was the highest [20]. However, the results of these studies would not necessarily translate to an older ambulatory elderly population. The latter type of data would be necessary to inform our health officials to consider the timing for lifting of restrictions and to inform the future needs for masking and other public health measures.

Traditionally clinic trials are conducted with onsite based approaches as this allows researchers to closely monitor each study participant in a controlled setting [21]. The Covid-19 pandemic highlighted the limitations of centralized clinical trials. Public health restrictions forced hospital and doctors office-based study coordination centers to temporarily close their sites for any non-essential in-person research visits, disrupting enrollment and follow up research activities [22]. In response, many centralized clinical trials adapted their research methods to enable virtual and electronic data collection.

Given these limitations in conducting research, we designed the STOPCoV study (Safety and Efficacy of Preventative COVID Vaccines) in the midst of the global Covid-19 pandemic in order to assess the impact of age ($\geq$ 70 years vs. 30–50 years) on the antibody response and the safety of the approved vaccines over time (www.stopcov.ca) [23]. This clinical trial needed to be decentralized to meet public health restrictions, and have a flexible design and regulatory approval in order to adjust to changing vaccine recommendations from public health officials in Ontario [3]. The Data Aggregation, Translation and Architecture (DATA) and Healthcare Human Factors (HHF) teams at the University Health Network (UHN) collaborated with the study team to develop an online digital platform to enable decentralized study procedures such as recruitment, informed e-consent (with videos and interspersed questions to re-enforce comprehension), baseline and follow-up questionnaires, symptom diaries, monthly check-ins, video instructions, and email reminders for study activities. Other study procedures such as dried blood spots were completed via mail. With the Omicron wave of the epidemic, it became clear that breakthrough infection could occur despite vaccination. In response, we added a sub-study to our protocol to estimate the rate of, and factors associated with, breakthrough infections and the predictive value of the rapid antigen tests in diagnosing infection in a vaccinated population (4). Again, we needed to adapt our digital platform to enable consent, training of our participants and data capture for this sub-study.

This was a new research method for us and our institution. We had to rapidly develop our platform and make frequent changes due to the rapid changing recommendations of our government with respect to vaccine doses, brands and intervals. We anticipated that many of our participants had never been previously involved in research and may not have had much experience with digital platforms. Consequently, we designed this survey to determine the satisfaction of our participants with our protocol, tools and procedures. We hypothesized that age could have an impact on satisfaction given the need for some digital proficiency and confidence to participate. Our aim was to identify areas to improve the process and procedures and share our experience to inform future fully decentralized research studies.

## Methods

Full methods for the main STOPCoV study [23] and the rapid antigen test sub-study [24] are available but are described briefly below.

### Design

A longitudinal cohort study planned to follow participants with two COVID-19 vaccine doses for 48 weeks. Trial registration: Clinicaltrials.gov. NCT05208983. Participants were eligible if they were between 30–50 years or $\geq$70 years and received their vaccine through an Ontario Distribution center. We purposely planned for 70% enrollment in the older cohort.

### Study region

Participants were recruited from across the province of Ontario, Canada. According to Census Canada 2021, there are 14, 223, 942 persons living in Ontario, with a total area of 15.2/km2, and a density of 1 million persons/km2. Approximately 2/3 are in urban areas with Toronto representing the largest metropolitan area, with a population of 6.6M. Of the Ontario population, 95% are English speaking, 34% identify as a visible minority and 5.5% as Black. Of the Ontario population, Of the population 12% are men and 14% women between the ages of 30–50, and 4.9% men and 5.7% women between the ages of 70 and 85 years.

## Recruitment

A data sharing agreement with the Ontario Ministry of Health enabled us to send study information emails to persons receiving the COVID-19 vaccine at an Ontario distribution center who consented to contact for research. A similar email was sent to Ontario Canadian Association of Retired Persons members (www.carp.ca). Participants could enrol through the website prior to the first or second vaccine dose that had to be administered in Ontario. A total of 1286 adults (911 older and 375 younger) self-recruited between May 17- July 31, 2021. Five participants did not meet eligibility criteria and at the time of initiation of the rapid antigen sub-study, 20 had withdrawn consent, leaving 1261 (98%) continuing in the cohort. These active participants were sent an email describing the sub-study in mid-January 2022. 806 participants e-consented and 727 (90%) completed > 1 rapid antigen test (RAT) January 28- March 29, 2022.

**Electronic consent** of the initial and substudy including the request to share core data elements with the Canadian Immunity Task Force was completed on the study website. The study and electronic consent process were approved by the University Health Network (UHN) Ethics Review Committee. Consented participants used the study website with their personal identification (ID) number and password as a portal for communication with study staff, data collection and results reporting. A schedule for required activities and email reminders were provided.

## Questionnaires

Self-administered electronic questionnaires collected baseline demographic and health data. Racial background was self reported. Electronic diaries collected data on vaccine date and brand and local and systemic adverse events for 7-days after the first three vaccine doses. Monthly check-in questionnaires capture persistent vaccine related adverse events, booster doses and new COVID-19 diagnoses.

## Dried Blood Spot (DBS) Specimens

Samples were requested +/-7 days of initial vaccine, three weeks (+/-1 week) after the first vaccine dose, two weeks (+/-1 week) after the second vaccine dose and then every 12 weeks (+/-3 weeks). If the dose interval exceeded 28 days, an additional sample was collected prior to the second vaccine dose. Additional DBS were requested 3–4 weeks after vaccine boosters. DBS collected on Whatman 903 cards using a lancet for finger-prick.

## Satisfaction survey

The participant feedback survey was modelled after the 'Study Participant Feedback Questionnaire Toolkit' created by TransCelerate Biopharma [25] and included questions related to each phase of the STOPCoV study and the rapid antigen test sub-study, including recruitment, consent process, study procedure, digital platform and interaction with the study team. The questions were designed to be answered on a 5-point Likert scale. After an initial set of questions was developed, 3 members of our research team met in person to determine by consensus which questions should be included and to ensure they were comprehensive and clear. A final set of 42 questions that could be answered in under 15 minutes was formed and tested with three non-study participants in the same age range as our participants to obtain feedback on clarity and ease of use. REDCap, an external secure web platform, was used to distribute the survey via email to active participants and collect responses. The survey was reviewed and approved by the University Health Network Research Ethics Board.

Demographic and clinical characteristics of participants who responded to the survey were compared to those who did not using $\chi^2$-tests for categorical variables and Wilcoxon Rank Sum tests for continuous variables. Only 3 individuals reported their gender as non-binary, they have been included in the female category for analysis. Individual survey item responses were summarized by age cohort using counts and percentages, and comparisons of the items were made between age cohorts using $\chi^2$-tests for trend. For the comparisons, non-responses were included as separate categories, and for the Likert-type questions with five possible categories, we formed two categories by combining the top two levels and combining the bottom three levels and non-responses (e.g. "Strongly Agree" or "Agree" were compared to "Neither Agree nor Disagree", "Disagree", "Strongly Disagree", or "Not Answered"); an $\alpha$-level of 0.05 was used.

## Results

The survey link with an introduction to the purpose of this sub-study was sent by email to 1253 active participants near the mid-way point of the main STOPCoV trial (April 2022). They were informed that they would not be identified by name and were not required to answer all questions. Of these, 726 had also participated in the rapid antigen test sub-study. In total 883 (70%) responded to the survey including 701 (83%) from the older and 182 (54%) from the younger cohort, with no difference by sex. We sent one reminder to those who did not complete the survey after 8 days. We closed the survey 14 days after the first email inviting participation. Table 1 compares the self- reported demographics of those who responded vs those who did not respond to the survey. While all of the participants who did the survey also completed the baseline questionnaire for the main study only 80.3% of the people who did not do the survey had baseline data available (p<0.0001). For the participants who did have baseline data: those who did the survey were more likely to be from the older cohort (79% vs. 49% p<0.0001), white (90% vs. 82%, p = 0.0015), previous smokers (40% vs. 31% p<0.0001), and have underlying cardiovascular disease (39% vs. 28%, p = 0.0006) or a cancer diagnosis (17% vs. 9%, p = 0.00052) compared to those who did not complete the survey. Gender, other comorbidities, were not different by survey participation.

Table 2 shows the responses to items regarding clinical research experience. Prior to the STOPCoV study, only 33% of both male and female participants had participated in a clinical research study more in the older cohort. However, 47% of the participants agreed that the Covid-19 pandemic changed their awareness of clinical research studies. 95% of participants were satisfied to very satisfied with respect to their participation in the STOPCoV clinical trial and 90% indicated they would participate in research again. Responses to these questions did not vary by age category or sex.

Recruitment strategies, reasons for participating in the main STOPCoV study, and questions about the e-consent process are shown in Table 3. Recruitment in the older cohort was mainly from response to an email sent from the study team (40%), or a targeted email from a seniors magazine mailing list (ZOOMER/CARP) (14%) or referral from a friend or relative (13%). The younger cohort were most likely to have responded to an email from the study team (36%) or from a referral for a friend or relative (28%). The younger cohort (especially the females) were more likely to have enrolled based on learning about the study through social media like Facebook or twitter 19%) compared to 2% of the older cohort.

The leading reasons participants decided to enroll were to help others learn more about the antibody response to the COVID vaccine (76% overall, 90% younger, 72% older cohort, p < .001)), to learn about their own personal antibody response (68%, overall, 84% younger, 65% older cohort, p < .001) especially the females (72% vs 61% males, p = .001), to contribute to

**Table 1. Demographics by Survey Participation.**

| | Did not do the survey | Did the survey | p |
|---|---|---|---|
| n | 297[a] | 883 | |
| Age 70+ Cohort | 144 (48.5%) | 701 (79.4%) | <0.0001 |
| Female or Non-Binary | 193 (65%) | 565 (64%) | 0.81 |
| **Racial Background (self reported)** | | | 0.0015 |
| Arab/West Indian | 5 (1.7%) | 5 (0.6%) | |
| Black | 8 (2.7%) | 11 (1.2%) | |
| Indigenous/Aboriginal/Indian or Native American | 0 (0%) | 5 (0.6%) | |
| Latin-American | 5 (1.7%) | 2 (0.2%) | |
| South Asian | 8 (2.7%) | 7 (0.8%) | |
| Southeast Asian | 11 (3.7%) | 21 (2.4%) | |
| White | 242 (81.5%) | 798 (90.4%) | |
| Other | 18 (6.1%) | 34 (3.9%) | |
| **Smoking Status** | | | <0.0001 |
| Never | 173 (58.2%) | 493 (55.8%) | |
| Previous | 92 (31.0%) | 357 (40.4%) | |
| Current | 32 (10.8%) | 33 (3.7%) | |
| **Comorbidities** | | | |
| Diabetes | 32 (10.8%) | 91 (10.3%) | 0.83 |
| Cardiovascular Disease | 82 (27.6%) | 341 (38.6%) | 0.00060 |
| Cancer | 27 (9.1%) | 152 (17.2%) | 0.00052 |
| Transplant or Immunosuppressed | 10 (3.4%) | 37 (4.2%) | 0.61 |
| Chronic Obstructive Lung Disease | 5 (1.7%) | 17 (1.9%) | 0.99 |
| Asthma | 35 (11.8%) | 87 (9.9%) | 0.38 |
| Chronic Kidney Disease | 4 (1.3%) | 15 (1.7%) | 0.80 |
| Hepatitis C | 3 (1.0%) | 2 (0.2%) | 0.10 |
| Chronic Liver Disease | 4 (1.3%) | 8 (0.9%) | 0.51 |
| Chronic Blood Disease | 2 (0.7%) | 11 (1.2%) | 0.54 |
| Chronic Neurologic Disease | 3 (1.0%) | 15 (1.7%) | 0.59 |
| Dialysis | 1 (0.3%) | 5 (0.6%) | 0.99 |

a 73/370 (19.7%) participants who were invited but did not complete the survey did not have demographic data available.

knowledge about the safety of COVID vaccines (73%) as well as the convenience of taking part in a clinical trial from home (37%). The latter two factors were more commonly reported by the females.

Over 90% of participants (no difference by age group or sex) responded that they understood what study procedures were expected of them before consenting to participate. Over 60% of participants (67% of the older cohort, 60% younger cohort, p.005); no difference by sex watched the consent video and 65% of the older and 58% of the younger cohort, p.085) found the explanation of the consent process helpful.

All reporting for the main study was done through the STOPCoV website; satisfaction questions about the study website and the schedule of the study are shown in Table 4. Overall, 90% of participants in both sex and age categories agreed/strongly agreed the study website was easy to use. Both age cohorts agreed/strongly agreed that it was easy to log in and participate in the study using their electronic device. Participants reported that the monthly questionnaires were easy to understand and they were able to navigate through the website with ease to record results and symptoms in their profile. Over 80% of participants in both age and sex cohorts

**Table 2. Experience with Research.**

| | Total (n = 883) | Age 30–50 (n = 182) | Age 70 and over (n = 701) | P-value | Male (n = 318) | Female or non-binary (n = 565) | P-value |
|---|---|---|---|---|---|---|---|
| **Have you ever participated in a Clinical Research study prior to the STOPCoV study?[a]** | 294 (33.3%) | 38 (20.9%) | 256 (36.5%) | <0.001* | 103 (32.4%) | 191 (33.8%) | 0.16* |
| **Was it** | | | | <0.001* | | | 0.36* |
| Once before | 159 (18.0%) | 15 (8.2%) | 144 (20.5%) | | 62 (19.5%) | 97 (17.2%) | |
| More than once before | 132 (14.9%) | 23 (12.6%) | 109 (15.5%) | | 41 (12.9%) | 91 (16.1%) | |
| Not Answered | 592 (67.0%) | 144 (79.1%) | 448 (63.9%) | | 215 (67.6%) | 377 (66.7%) | |
| **I think clinical research studies are important** | | | | 0.11† | | | 0.80† |
| Strongly Agree | 769 (87.1%) | 169 (92.9%) | 600 (85.6%) | | 273 (85.8%) | 496 (87.8%) | |
| Agree | 100 (11.3%) | 13 (7.1%) | 87 (12.4%) | | 39 (12.3%) | 61 (10.8%) | |
| Neither Agree nor Disagree | 4 (0.5%) | 0 (0%) | 4 (0.6%) | | 3 (0.9%) | 1 (0.2%) | |
| Disagree | 0 (0%) | 0 (0%) | 0 (0%) | | 0 (0%) | 0 (0%) | |
| Strongly Disagree | 3 (0.3%) | 0 (0%) | 3 (0.4%) | | 0 (0.0%) | 3 (0.5%) | |
| Not Answered | 7 (0.8%) | 0 (0%) | 7 (1.0%) | | 3 (0.9%) | 4 (0.7%) | |
| **The COVID pandemic changed my awareness of clinical research studies** | | | | 0.16† | | | 0.35† |
| Strongly Agree | 175 (19.8%) | 44 (24.2%) | 131 (18.7%) | | 55 (17.3%) | 120 (21.2%) | |
| Agree | 242 (27.4%) | 33 (18.1%) | 209 (29.8%) | | 88 (27.7%) | 154 (27.3%) | |
| Neither Agree nor Disagree | 336 (38.1%) | 68 (37.4%) | 268 (38.2%) | | 131 (41.2%) | 205 (36.3%) | |
| Disagree | 74 (8.4%) | 25 (13.7%) | 49 (7.0%) | | 22 (6.9%) | 52 (9.2%) | |
| Strongly Disagree | 38 (4.3%) | 12 (6.6%) | 26 (3.7%) | | 15 (4.7%) | 23 (4.1%) | |
| Not Answered | 18 (2.0%) | 0 (0%) | 18 (2.6%) | | 7 (2.2%) | 11 (1.9%) | |
| **Overall, I was satisfied with my study experience** | | | | 0.83† | | | 0.96† |
| Strongly Agree | 526 (59.6%) | 122 (67.0%) | 404 (57.6%) | | 173 (54.4%) | 353 (62.5%) | |
| Agree | 308 (34.9%) | 51 (28.0%) | 257 (36.7%) | | 128 (40.3%) | 180 (31.9%) | |
| Neither Agree nor Disagree | 22 (2.5%) | 2 (1.1%) | 20 (2.9%) | | 11 (3.5%) | 11 (1.9%) | |
| Disagree | 9 (1.0%) | 0 (0%) | 9 (1.3%) | | 2 (0.6%) | 7 (1.2%) | |
| Strongly Disagree | 5 (0.6%) | 2 (1.1%) | 3 (0.4%) | | 3 (0.9%) | 2 (0.4%) | |
| Not Answered | 13 (1.5%) | 5 (2.7%) | 8 (1.1%) | | 1 (0.3%) | 12 (2.1%) | |
| **I would be willing to participate in clinical research** | | | | 0.099† | | | 0.14† |
| Strongly Agree | 496 (56.2%) | 112 (61.5%) | 384 (54.8%) | | 155 (48.7%) | 341 (60.4%) | |
| Agree | 303 (34.3%) | 59 (32.4%) | 244 (34.8%) | | 126 (39.6%) | 177 (31.3%) | |
| Neither Agree nor Disagree | 57 (6.5%) | 5 (2.7%) | 52 (7.4%) | | 30 (9.4%) | 27 (4.8%) | |
| Disagree | 6 (0.7%) | 0 (0%) | 6 (0.9%) | | 3 (0.9%) | 3 (0.5%) | |
| Strongly Disagree | 3 (0.3%) | 1 (0.5%) | 2 (0.3%) | | 1 (0.3%) | 2 (0.4%) | |

(*Continued*)

**Table 2.** (Continued)

| | Total (n = 883) | Age 30–50 (n = 182) | Age 70 and over (n = 701) | P-value | Male (n = 318) | Female or non-binary (n = 565) | P-value |
|---|---|---|---|---|---|---|---|
| Not Answered | 18 (2.0%) | 5 (2.7%) | 13 (1.9%) | | 3 (0.9%) | 15 (2.7%) | |

[a] 9 participants aged 70 and older did not answer this question

\* Chi-squared test for trend (exact test used when expected cell counts were low)

†Chi-squared test (2X2) for dichotomized response categories: "Strongly Agree" or "Agree" vs. others (including "Not Answered")

strongly agreed that the email reminders that were sent when monthly questionnaires and DBS were due were very helpful. Whenever changes were made to the design of the study website to collect new information, the site needed to be refreshed/cache cleared; 12% of participants reported that they had difficulty doing this more commonly in the younger cohort 17% vs 11% of the older cohort, p = .027) and in the women 4.2% compared to the men 3.5%, p = .012. Over 80% of participants (no difference by age or sex category) agreed that compared to when the study started, the overall commitment required was close to what they had expected.

Table 5 shows responses to questions regarding dried blood spot collection and rapid antigen tests. Initially, there was some confusion about how to use the lancets to collect the dry blood spot (DBS) and many participants called in for instruction. Although not available at the study start, 70% of the study participants subsequently watched the video explaining the dried blood spot collection procedure and found the instructional video helpful. While 78% of respondents answered there was little to no difficulty using the lancets and collecting their DBS (the men reported slightly more problems, p = .027), 39% did report difficulty getting sufficient blood to put on the DBS collection card at least some of the time, but this did not differ by age or sex category. Only a small per cent reported unacceptable discomfort with the blood prick. Over 90% in both age and sex cohorts found the frequency of DBS collection was acceptable. Over 75% of participants felt that their antibody results were presented in a way that was easy to understand (no sex difference) but this was less common in the older 78% than the younger cohort 85%, p = .036).

An instructional video was available at the time of recruitment to the Rapid Antigen Test (RAT) sub-study. 41% of the participants (43% older, 36% younger cohort, p < .001) reported watching this video with 41% finding it helpful, and 69% older and 76% younger, p = .097) participants felt confident interpreting the results of the RAT with no difference by sex.

Interactions with study team are described in Table 6. A dedicated study email and phone line for contacting the study team was available to address concerns and provide assistance. The study team was contacted frequently by phone 38% and by email 66%. Overall, the older cohort was more likely to contact the study team by phone 44% vs 18% of the younger cohort, p = .001. In contrast the younger cohort were more likely to contact the study team by email (74% vs 64%, p = .01) Females were more likely to use email 69% than men 60%, p < .01. Participants in both age cohorts (especially the females) answered that the study staff was both easy to reach (80%) and treated them with respect (90%). 79% of both age cohorts especially the females (85% vs 74% men, p < .001) were satisfied with the answers to their questions.

## Discussion

Overall, the responses to the feedback survey were positive and 95% of participants in both age and sex cohorts were satisfied to very satisfied with respect to their participation in the STOP-CoV clinical trial. This is further exemplified by the high proportion of respondents to this

**Table 3. Recruitment and E-Consent.**

| | Total (n = 883) | Age 30–50 (n = 182) | Age 70 and over (n = 701) | P-value | Male (n = 318) | Female or non-binary (n = 565) | P-value |
|---|---|---|---|---|---|---|---|
| **How did you learn about the STOPCoV Study?** | | | | <0.001* | | | <0.001* |
| Flyer/Poster/Advertisement | 75 (8.5%) | 7 (3.8%) | 68 (9.7%) | | 21 (6.6%) | 54 (9.6%) | |
| Email from the study team | 348 (39.4%) | 65 (35.7%) | 283 (40.4%) | | 136 (42.8%) | 212 (37.5%) | |
| Email from Zoomer/CARP | 101 (11.4%) | 0 (0%) | 101 (14.4%) | | 29 (9.1%) | 72 (12.7%) | |
| Friend/Relative | 142 (16.1%) | 51 (28.0%) | 91 (13.0%) | | 52 (16.4%) | 90 (15.9%) | |
| Facebook/Twitter | 48 (5.4%) | 34 (18.7%) | 14 (2.0%) | | 7 (2.2%) | 41 (7.3%) | |
| I don't remember | 138 (15.6%) | 23 (12.6%) | 115 (16.4%) | | 65 (20.4%) | 73 (12.9%) | |
| Not Answered | 31 (3.5%) | 2 (1.1%) | 29 (4.1%) | | 8 (2.5%) | 23 (4.1%) | |
| **Reasons for participation in STOPCov Study[a]** | | | | | | | |
| Chance to learn about my personal antibody response to the COVID vaccine | 604 (68.4%) | 152 (83.5%) | 452 (64.5%) | <0.001 | 195 (61.3%) | 409 (72.4%) | 0.001 |
| Chance to help others learn more about the antibody response to the COVID vaccine | 668 (75.7%) | 163 (89.6%) | 505 (72.0%) | <0.001 | 233 (73.3%) | 435 (77.0%) | 0.25 |
| Study was recommended to me | 66 (7.5%) | 11 (6.0%) | 55 (7.8%) | 0.51 | 26 (8.2%) | 40 (7.1%) | 0.64 |
| I could participate from my home | 329 (37.3%) | 75 (41.2%) | 254 (36.2%) | 0.25 | 99 (31.1%) | 230 (40.7%) | 0.006 |
| To contribute to knowledge about the safety of the COVID vaccine | 641 (72.6%) | 138 (75.8%) | 503 (71.8%) | 0.32 | 214 (67.3%) | 427 (75.6%) | 0.01 |
| **Did you view the video explaining the Consent Process?** | | | | 0.005* | | | 0.18* |
| Yes | 575 (65.1%) | 105 (57.7%) | 470 (67.0%) | | 195 (61.3%) | 380 (67.3%) | |
| No | 288 (32.6%) | 76 (41.8%) | 212 (30.2%) | | 116 (36.5%) | 172 (30.4%) | |
| Not Answered | 20 (2.3%) | 1 (0.5%) | 19 (2.7%) | | 7 (2.2%) | 13 (2.3%) | |
| **Did you find the explanation of the consent process helpful?** | | | | 0.085* | | | 0.14* |
| Yes | 562 (63.6%) | 105 (57.7%) | 457 (65.2%) | | 190 (59.7%) | 372 (65.8%) | |
| No | 4 (0.5%) | 0 (0%) | 4 (0.6%) | | 1 (0.3%) | 3 (0.5%) | |
| Not Answered | 317 (35.9%) | 77 (42.3%) | 240 (34.2%) | | 127 (39.9%) | 190 (33.6%) | |
| **I understood what study procedures were expected of me before consenting to participate in the study** | | | | 0.11† | | | 0.57† |
| Strongly Agree | 521 (59.0%) | 143 (78.6%) | 378 (53.9%) | | 167 (52.5%) | 354 (62.7%) | |
| Agree | 308 (34.9%) | 33 (18.1%) | 275 (39.2%) | | 134 (42.1%) | 174 (30.8%) | |
| Neither Agree nor Disagree | 28 (3.2%) | 4 (2.2%) | 24 (3.4%) | | 7 (2.2%) | 21 (3.7%) | |
| Disagree | 12 (1.4%) | 0 (0%) | 12 (1.7%) | | 6 (1.9%) | 6 (1.1%) | |
| Strongly Disagree | 3 (0.3%) | 1 (0.5%) | 2 (0.3%) | | 1 (0.3%) | 2 (0.4%) | |
| Not Answered | 11 (1.2%) | 1 (0.5%) | 10 (1.4%) | | 3 (0.9%) | 8 (1.4%) | |

CARP, Canadian Association of Retired Persons

a Participants can select multiple reasons

*Chi-squared test for trend (exact test used when expected cell counts were low)

†Chi-squared test for dichotomized response categories: "Strongly Agree" or "Agree" vs. others (including "Not Answered")

**Table 4. Study Website and Schedule.**

| | Total (n = 883) | Age 30–50 (n = 182) | Age 70 and over (n = 701) | P-value | Male (n = 318) | Female or non-binary (n = 565) | P-value |
|---|---|---|---|---|---|---|---|
| **I found the study website ([www.stopcov.ca](www.stopcov.ca)) easy to use** | | | | 0.39† | | | 0.98† |
| Strongly Agree | 496 (56.2%) | 111 (61.0%) | 385 (54.9%) | | 164 (51.6%) | 332 (58.8%) | |
| Agree | 302 (34.2%) | 57 (31.3%) | 245 (35.0%) | | 124 (39.0%) | 178 (31.5%) | |
| Neither Agree nor Disagree | 60 (6.8%) | 10 (5.5%) | 50 (7.1%) | | 25 (7.9%) | 35 (6.2%) | |
| Disagree | 12 (1.4%) | 2 (1.1%) | 10 (1.4%) | | 3 (0.9%) | 9 (1.6%) | |
| Strongly  Disagree | 2 (0.2%) | 0 (0%) | 2 (0.3%) | | 1 (0.3%) | 1 (0.2%) | |
| Not Answered | 11 (1.2%) | 2 (1.1%) | 9 (1.3%) | | 1 (0.3%) | 10 (1.8%) | |
| **It was easy participating in the study using my electronic device** | | | | 0.50† | | | 0.47† |
| Strongly Agree | 585 (66.3%) | 136 (74.7%) | 449 (64.1%) | | 200 (62.9%) | 385 (68.1%) | |
| Agree | 253 (28.7%) | 39 (21.4%) | 214 (30.5%) | | 99 (31.1%) | 154 (27.3%) | |
| Neither Agree nor Disagree | 27 (3.1%) | 3 (1.6%) | 24 (3.4%) | | 13 (4.1%) | 14 (2.5%) | |
| Disagree | 6 (0.7%) | 1 (0.5%) | 5 (0.7%) | | 2 (0.6%) | 4 (0.7%) | |
| Strongly Disagree | 3 (0.3%) | 1 (0.5%) | 2 (0.3%) | | 1 (0.3%) | 2 (0.4%) | |
| Not Answered | 9 (1.0%) | 2 (1.1%) | 7 (1.0%) | | 3 (0.9%) | 6 (1.1%) | |
| **It was easy for me to log into my study portal and update my information** | | | | 0.82† | | | 0.93† |
| Strongly Agree | 577 (65.3%) | 129 (70.9%) | 448 (63.9%) | | 201 (63.2%) | 376 (66.5%) | |
| Agree | 237 (26.8%) | 40 (22.0%) | 197 (28.1%) | | 93 (29.2%) | 144 (25.5%) | |
| Neither Agree nor Disagree | 40 (4.5%) | 7 (3.8%) | 33 (4.7%) | | 14 (4.4%) | 26 (4.6%) | |
| Disagree | 20 (2.3%) | 3 (1.6%) | 17 (2.4%) | | 9 (2.8%) | 11 (1.9%) | |
| Strongly Disagree | 1 (0.1%) | 0 (0%) | 1 (0.1%) | | 0 (0.0%) | 1 (0.2%) | |
| Not Answered | 8 (0.9%) | 3 (1.6%) | 5 (0.7%) | | 1 (0.3%) | 7 (1.2%) | |
| **The questionnaires were easy to understand and complete** | | | | 0.82† | | | 0.22† |
| Strongly Agree | 599 (67.8%) | 136 (74.7%) | 463 (66.0%) | | 196 (61.6%) | 403 (71.3%) | |
| Agree | 260 (29.4%) | 42 (23.1%) | 218 (31.1%) | | 110 (34.6%) | 150 (26.5%) | |
| Neither Agree nor Disagree | 13 (1.5%) | 2 (1.1%) | 11 (1.6%) | | 7 (2.2%) | 6 (1.1%) | |
| Disagree | 1 (0.1%) | 0 (0%) | 1 (0.1%) | | 1 (0.3%) | 0 (0.0%) | |
| Strongly Disagree | 0 (0%) | 0 (0%) | 0 (0%) | | 0 (0%) | 0 (0%) | |
| Not Answered | 10 (1.1%) | 2 (1.1%) | 8 (1.1%) | | 4 (1.3%) | 6 (1.1%) | |
| **I was able to record my results and symptoms easily in my site profile** | | | | 0.34† | | | 0.17† |
| Strongly Agree | 556 (63.0%) | 132 (72.5%) | 424 (60.5%) | | 185 (58.2%) | 371 (65.7%) | |
| Agree | 267 (30.2%) | 41 (22.5%) | 226 (32.2%) | | 106 (33.3%) | 161 (28.5%) | |
| Neither Agree nor Disagree | 36 (4.1%) | 6 (3.3%) | 30 (4.3%) | | 18 (5.7%) | 18 (3.2%) | |
| Disagree | 9 (1.0%) | 1 (0.5%) | 8 (1.1%) | | 5 (1.6%) | 4 (0.7%) | |
| Strongly Disagree | 1 (0.1%) | 0 (0%) | 1 (0.1%) | | 0 (0.0%) | 1 (0.2%) | |
| Not Answered | 14 (1.6%) | 2 (1.1%) | 12 (1.7%) | | 4 (1.3%) | 10 (1.8%) | |
| **I had difficulty refreshing/caching my page each time there was a study update** | | | | 0.027† | | | 0.012† |
| Strongly Agree | 35 (4.0%) | 14 (7.7%) | 21 (3.0%) | | 11 (3.5%) | 24 (4.2%) | |

*(Continued)*

**Table 4.** (Continued)

| | Total (n = 883) | Age 30–50 (n = 182) | Age 70 and over (n = 701) | P-value | Male (n = 318) | Female or non-binary (n = 565) | P-value |
|---|---|---|---|---|---|---|---|
| Agree | 71 (8.0%) | 17 (9.3%) | 54 (7.7%) | | 15 (4.7%) | 56 (9.9%) | |
| Neither Agree nor Disagree | 155 (17.6%) | 27 (14.8%) | 128 (18.3%) | | 54 (17.0%) | 101 (17.9%) | |
| Disagree | 383 (43.4%) | 66 (36.3%) | 317 (45.2%) | | 149 (46.9%) | 234 (41.4%) | |
| Strongly Disagree | 223 (25.3%) | 56 (30.8%) | 167 (23.8%) | | 84 (26.4%) | 139 (24.6%) | |
| Not Answered | 16 (1.8%) | 2 (1.1%) | 14 (2.0%) | | 5 (1.6%) | 11 (1.9%) | |
| **The email reminders were helpful** | | | | >0.99† | | | 0.28† |
| Strongly Agree | 735 (83.2%) | 159 (87.4%) | 576 (82.2%) | | 249 (78.3%) | 486 (86.0%) | |
| Agree | 133 (15.1%) | 20 (11.0%) | 113 (16.1%) | | 66 (20.8%) | 67 (11.9%) | |
| Neither Agree nor Disagree | 5 (0.6%) | 0 (0%) | 5 (0.7%) | | 1 (0.3%) | 4 (0.7%) | |
| Disagree | 1 (0.1%) | 1 (0.5%) | 0 (0%) | | 0 (0.0%) | 1 (0.2%) | |
| Strongly Disagree | 0 (0%) | 0 (0%) | 0 (0%) | | 0 (0.0%) | 0 (0%) | |
| Not Answered | 9 (1.0%) | 2 (1.1%) | 7 (1.0%) | | 2 (0.6%) | 7 (1.2%) | |
| **The schedule of expected study activities was easy to follow** | | | | 0.092† | | | 0.055† |
| Strongly Agree | 427 (48.4%) | 86 (47.3%) | 341 (48.6%) | | 137 (43.1%) | 290 (51.3%) | |
| Agree | 354 (40.1%) | 68 (37.4%) | 286 (40.8%) | | 135 (42.5%) | 219 (38.8%) | |
| Neither Agree nor Disagree | 59 (6.7%) | 11 (6.0%) | 48 (6.8%) | | 32 (10.1%) | 27 (4.8%) | |
| Disagree | 30 (3.4%) | 15 (8.2%) | 15 (2.1%) | | 10 (3.1%) | 20 (3.5%) | |
| Strongly Disagree | 1 (0.1%) | 0 (0%) | 1 (0.1%) | | 0 (0.0%) | 1 (0.2%) | |
| Not Answered | 12 (1.4%) | 2 (1.1%) | 10 (1.4%) | | 4 (1.3%) | 8 (1.4%) | |
| **Compared to when the study started, the overall commitment required was close to what I expected** | | | | >0.99† | | | 0.17† |
| Strongly Agree | 393 (44.5%) | 90 (49.5%) | 303 (43.2%) | | 125 (39.3%) | 268 (47.4%) | |
| Agree | 371 (42.0%) | 67 (36.8%) | 304 (43.4%) | | 143 (45.0%) | 228 (40.4%) | |
| Neither Agree nor Disagree | 88 (10.0%) | 13 (7.1%) | 75 (10.7%) | | 42 (13.2%) | 46 (8.1%) | |
| Disagree | 17 (1.9%) | 5 (2.7%) | 12 (1.7%) | | 6 (1.9%) | 11 (1.9%) | |
| Strongly Disagree | 5 (0.6%) | 2 (1.1%) | 3 (0.4%) | | 2 (0.6%) | 3 (0.5%) | |
| Not Answered | 9 (1.0%) | 5 (2.7%) | 4 (0.6%) | | 0 (0.0%) | 9 (1.6%) | |

†Chi-squared test for dichotomized response categories: "Strongly Agree" or "Agree" vs. others (including "Not Answered")

satisfaction survey. The older cohort and those with underlying cardiovascular disease or cancer were more likely to respond to the survey which is consistent with their overall study participation and may reflect concern for their own health.

Main reasons for participation in the STOPCoV study were to learn about their own personal antibody response (68%) as well as help others learn more about the antibody response to the COVID vaccine (76%). This was an important observation as many research ethics boards/researchers do not allow research results to be given directly to participants as their significance may be unclear and misinterpreted [26,27]. However, this was an evolving pandemic and providing research participants with their antibody levels at multiple time points helped

**Table 5. Dried Blood Spots and Rapid Antigen Tests.**

| | Total (n = 883) | Age 30–50 (n = 182) | Age 70 and over (n = 701) | P-value | Male (n = 318) | Female or non-binary (n = 565) | P-value |
|---|---|---|---|---|---|---|---|
| **Did you view the video explaining how to collect your Dry Blood Spot Specimen?** | | | | 0.14* | | | 0.26* |
| Yes | 632 (71.6%) | 125 (68.7%) | 507 (72.3%) | | 217 (68.2%) | 415 (73.5%) | |
| No | 222 (25.1%) | 54 (29.7%) | 168 (24.0%) | | 89 (28.0%) | 133 (23.5%) | |
| Not Answered | 29 (3.3%) | 3 (1.6%) | 26 (3.7%) | | 12 (3.8%) | 17 (3.0%) | |
| **Did you find Dry Blood Spot instructional video helpful?** | | | | 0.56* | | | 0.26* |
| Yes | 616 (69.8%) | 124 (68.1%) | 492 (70.2%) | | 211 (66.4%) | 405 (71.7%) | |
| No | 10 (1.1%) | 1 (0.5%) | 9 (1.3%) | | 4 (1.3%) | 6 (1.1%) | |
| Not Answered | 257 (29.1%) | 57 (31.3%) | 200 (28.5%) | | 103 (32.4%) | 154 (27.3%) | |
| **I felt confident on how to collect the Dry Blood Spot Specimen and mail it to the lab** | | | | 0.48† | | | 0.85† |
| All of the time | 579 (65.6%) | 121 (66.5%) | 458 (65.3%) | | 207 (65.1%) | 372 (65.8%) | |
| Most of the time | 227 (25.7%) | 48 (26.4%) | 179 (25.5%) | | 82 (25.8%) | 145 (25.7%) | |
| Some of the time | 26 (2.9%) | 3 (1.6%) | 23 (3.3%) | | 12 (3.8%) | 14 (2.5%) | |
| A little of the time | 10 (1.1%) | 3 (1.6%) | 7 (1.0%) | | 3 (0.9%) | 7 (1.2%) | |
| None of the time | 7 (0.8%) | 2 (1.1%) | 5 (0.7%) | | 4 (1.3%) | 3 (0.5%) | |
| Not Answered | 34 (3.9%) | 5 (2.7%) | 29 (4.1%) | | 10 (3.1%) | 24 (4.2%) | |
| **I had difficulty using the lancets (device supplied by the study to prick finger)** | | | | 0.18† | | | 0.027† |
| All of the time | 27 (3.1%) | 4 (2.2%) | 23 (3.3%) | | 14 (4.4%) | 13 (2.3%) | |
| Most of the time | 40 (4.5%) | 5 (2.7%) | 35 (5.0%) | | 19 (6.0%) | 21 (3.7%) | |
| Some of the time | 109 (12.3%) | 19 (10.4%) | 90 (12.8%) | | 32 (10.1%) | 77 (13.6%) | |
| A little of the time | 193 (21.9%) | 39 (21.4%) | 154 (22.0%) | | 64 (20.1%) | 129 (22.8%) | |
| None of the time | 498 (56.4%) | 110 (60.4%) | 388 (55.3%) | | 183 (57.5%) | 315 (55.8%) | |
| Not Answered | 16 (1.8%) | 5 (2.7%) | 11 (1.6%) | | 6 (1.9%) | 10 (1.8%) | |
| **I had difficulty in getting enough blood to put on the dried blood card** | | | | 0.42† | | | 0.85† |
| All of the time | 21 (2.4%) | 5 (2.7%) | 16 (2.3%) | | 9 (2.8%) | 12 (2.1%) | |
| Most of the time | 78 (8.8%) | 19 (10.4%) | 59 (8.4%) | | 28 (8.8%) | 50 (8.8%) | |
| Some of the time | 245 (27.7%) | 45 (24.7%) | 200 (28.5%) | | 85 (26.7%) | 160 (28.3%) | |
| A little of the time | 274 (31.0%) | 63 (34.6%) | 211 (30.1%) | | 101 (31.8%) | 173 (30.6%) | |
| None of the time | 255 (28.9%) | 47 (25.8%) | 208 (29.7%) | | 93 (29.2%) | 162 (28.7%) | |
| Not Answered | 10 (1.1%) | 3 (1.6%) | 7 (1.0%) | | 2 (0.6%) | 8 (1.4%) | |
| **The discomfort associated with pricking my finger was acceptable** | | | | 0.76† | | | 0.65† |
| All of the time | 605 (68.5%) | 108 (59.3%) | 497 (70.9%) | | 235 (73.9%) | 370 (65.5%) | |
| Most of the time | 165 (18.7%) | 49 (26.9%) | 116 (16.5%) | | 45 (14.2%) | 120 (21.2%) | |

(*Continued*)

**Table 5.** (Continued)

| | Total (n = 883) | Age 30–50 (n = 182) | Age 70 and over (n = 701) | P-value | Male (n = 318) | Female or non-binary (n = 565) | P-value |
|---|---|---|---|---|---|---|---|
| Some of the time | 22 (2.5%) | 7 (3.8%) | 15 (2.1%) | | 7 (2.2%) | 15 (2.7%) | |
| A little of the time | 25 (2.8%) | 6 (3.3%) | 19 (2.7%) | | 7 (2.2%) | 18 (3.2%) | |
| None of the time | 52 (5.9%) | 9 (4.9%) | 43 (6.1%) | | 21 (6.6%) | 31 (5.5%) | |
| Not Answered | 14 (1.6%) | 3 (1.6%) | 11 (1.6%) | | 3 (0.9%) | 11 (1.9%) | |
| **Did you choose to receive your antibody results?** | | | | 0.22* | | | 0.028* |
| Yes | 843 (95.5%) | 177 (97.3%) | 666 (95.0%) | | 299 (94.0%) | 544 (96.3%) | |
| No | 27 (3.1%) | 2 (1.1%) | 25 (3.6%) | | 16 (5.0%) | 11 (1.9%) | |
| Not Answered | 13 (1.5%) | 3 (1.6%) | 10 (1.4%) | | 3 (0.9%) | 10 (1.8%) | |
| **The antibody results presented in a way that was easy to understand** | | | | 0.036† | | | 0.94† |
| Strongly Agree | 341 (38.6%) | 94 (51.6%) | 247 (35.2%) | | 121 (38.1%) | 220 (38.9%) | |
| Agree | 359 (40.7%) | 61 (33.5%) | 298 (42.5%) | | 132 (41.5%) | 227 (40.2%) | |
| Neither Agree nor Disagree | 78 (8.8%) | 15 (8.2%) | 63 (9.0%) | | 28 (8.8%) | 50 (8.8%) | |
| Disagree | 46 (5.2%) | 6 (3.3%) | 40 (5.7%) | | 14 (4.4%) | 32 (5.7%) | |
| Strongly Disagree | 6 (0.7%) | 0 (0%) | 6 (0.9%) | | 1 (0.3%) | 5 (0.9%) | |
| Not Answered | 53 (6.0%) | 6 (3.3%) | 47 (6.7%) | | 22 (6.9%) | 31 (5.5%) | |
| **The frequency of Dry Blood Spot collection was** | | | | 0.29† | | | 0.62† |
| Too much | 4 (0.5%) | 2 (1.1%) | 2 (0.3%) | | 1 (0.3%) | 3 (0.5%) | |
| A little too much | 14 (1.6%) | 4 (2.2%) | 10 (1.4%) | | 4 (1.3%) | 10 (1.8%) | |
| Adequate | 818 (92.6%) | 162 (89.0%) | 656 (93.6%) | | 298 (93.7%) | 520 (92.0%) | |
| Not enough | 34 (3.9%) | 11 (6.0%) | 23 (3.3%) | | 12 (3.8%) | 22 (3.9%) | |
| Too little | 2 (0.2%) | 0 (0%) | 2 (0.3%) | | 1 (0.3%) | 1 (0.2%) | |
| Not Answered | 11 (1.2%) | 3 (1.6%) | 8 (1.1%) | | 2 (0.6%) | 9 (1.6%) | |
| **Did you view the video explaining how to collect your Rapid Antigen sample?** | | | | <0.001* | | | 0.23* |
| Yes | 369 (41.8%) | 65 (35.7%) | 304 (43.4%) | | 123 (38.7%) | 246 (43.5%) | |
| No | 262 (29.7%) | 76 (41.8%) | 186 (26.5%) | | 94 (29.6%) | 168 (29.7%) | |
| Not Answered | 252 (28.5%) | 41 (22.5%) | 211 (30.1%) | | 101 (31.8%) | 151 (26.7%) | |
| **Did you find the Rapid Antigen Test video helpful?** | | | | 0.20* | | | 0.083* |
| Yes | 361 (40.9%) | 64 (35.2%) | 297 (42.4%) | | 122 (38.4%) | 239 (42.3%) | |
| No | 6 (0.7%) | 1 (0.5%) | 5 (0.7%) | | 0 (0.0%) | 6 (1.1%) | |
| Not Answered | 516 (58.4%) | 117 (64.3%) | 399 (56.9%) | | 196 (61.6%) | 320 (56.6%) | |
| **I felt confident on how to collect the Rapid Antigen Sample** | | | | 0.12† | | | 0.35† |
| Strongly Agree | 384 (43.5%) | 91 (50.0%) | 293 (41.8%) | | 128 (40.3%) | 256 (45.3%) | |
| Agree | 237 (26.8%) | 46 (25.3%) | 191 (27.2%) | | 89 (28.0%) | 148 (26.2%) | |
| Neither Agree nor Disagree | 15 (1.7%) | 1 (0.5%) | 14 (2.0%) | | 4 (1.3%) | 11 (1.9%) | |
| Disagree | 4 (0.5%) | 3 (1.6%) | 1 (0.1%) | | 0 (0.0%) | 4 (0.7%) | |
| Strongly Disagree | 2 (0.2%) | 0 (0%) | 2 (0.3%) | | 0 (0.0%) | 2 (0.4%) | |

*(Continued)*

**Table 5.** (Continued)

| | Total (n = 883) | Age 30–50 (n = 182) | Age 70 and over (n = 701) | P-value | Male (n = 318) | Female or non-binary (n = 565) | P-value |
|---|---|---|---|---|---|---|---|
| Not Answered | 241 (27.3%) | 41 (22.5%) | 200 (28.5%) | | 97 (30.5%) | 144 (25.5%) | |
| **I felt confident interpreting the results of the Rapid Antigen test** | | | | 0.097† | | | 0.45† |
| Strongly Agree | 480 (54.4%) | 117 (64.3%) | 363 (51.8%) | | 165 (51.9%) | 315 (55.8%) | |
| Agree | 143 (16.2%) | 21 (11.5%) | 122 (17.4%) | | 54 (17.0%) | 89 (15.8%) | |
| Neither Agree nor Disagree | 12 (1.4%) | 3 (1.6%) | 9 (1.3%) | | 1 (0.3%) | 11 (1.9%) | |
| Disagree | 4 (0.5%) | 0 (0%) | 4 (0.6%) | | 1 (0.3%) | 3 (0.5%) | |
| Strongly Disagree | 1 (0.1%) | 0 (0%) | 1 (0.1%) | | 0 (0.0%) | 1 (0.2%) | |
| Not Answered | 243 (27.5%) | 41 (22.5%) | 202 (28.8%) | | 97 (30.5%) | 146 (25.8%) | |

*Chi-squared test for trend (exact test used when expected cell counts were low)

†Chi-squared test for dichotomized response categories

"Strongly Agree" or "Agree" vs. others (including "Not Answered")

"All of the time" or "Most of the time" vs. others (including "Not Answered")

"Too much" or "A little too much" vs. others (including "Not Answered")

with retention and to maintain participant engagement throughout the study. Participants were provided with a graph showing the results of individuals in their age cohort in relation to their individual values. A statement in the consent form and on the results page of the website reinforced that the results were research, the significance was unknown, and the team was available for questions. Although most reported in the survey that they were able to interpret their results many did contact the study team with questions of clarification.

Recruitment to the main STOPCoV study was rapid; 39.4% responded to an email sent by the study team based on their consent to be contacted for research at the time of their vaccination. The use of a senior's magazine mailing list (ZOOMER/CARP) to engage the older participants was successful with 11% reporting this as their primary contact. Another 5% responded to recruitment through social media (Facebook/Twitter) but this was much more successful with the younger and female participants where 19% reported this as the means of learning about the study. These responses highlight that recruitment strategies need to reflect the population of interest.

With the pandemic, electronic consents are increasingly being utilized, but there is controversy about whether they will be acceptable and understandable. [28] The majority of our participants (65%) especially the older group responded that they viewed the video posted on the STOPCoV website explaining the consent process and 64% found it helpful.

We had concerns about how easy it would be for our participants to use the study website to enter their data and that the older cohort might have trouble manipulating their portal. Furthermore, due to rapidly evolving vaccine recommendations and associated study procedures, the digital platform was not complete when the study was launched and every change required a page refresh and/or clearing cache which many of our participants were not familiar with. Despite these concerns, most respondents in both age and sex cohorts reported the study site easy to use suggesting that our participants were digitally proficient. We had anticipated that more of the older cohort would report caching issues but this was opposite to what was reported in the survey. This may have been a result of misinterpretation of the statement

**Table 6. Interaction with Study Staff.**

| | Total (n = 883) | Age 30–50 (n = 182) | Age 70 and over (n = 701) | P-value | Male (n = 318) | Female or non-binary (n = 565) | P-value |
|---|---|---|---|---|---|---|---|
| **The study staff was easy to reach** | | | | 0.63† | | | 0.003† |
| All of the time | 603 (68.3%) | 126 (69.2%) | 477 (68.0%) | | 194 (61.0%) | 409 (72.4%) | |
| Most of the time | 114 (12.9%) | 19 (10.4%) | 95 (13.6%) | | 47 (14.8%) | 67 (11.9%) | |
| Some of the time | 6 (0.7%) | 0 (0%) | 6 (0.9%) | | 0 (0.0%) | 6 (1.1%) | |
| A little of the time | 2 (0.2%) | 1 (0.5%) | 1 (0.1%) | | 0 (0.0%) | 2 (0.4%) | |
| None of the time | 1 (0.1%) | 0 (0%) | 1 (0.1%) | | 1 (0.3%) | 0 (0.0%) | |
| Not Answered | 157 (17.8%) | 36 (19.8%) | 121 (17.3%) | | 76 (23.9%) | 81 (14.3%) | |
| **Did you contact the study team by phone?** | | | | <0.001* | | | 0.66* |
| Yes | 338 (38.3%) | 33 (18.1%) | 305 (43.5%) | | 116 (36.5%) | 222 (39.3%) | |
| No | 532 (60.2%) | 144 (79.1%) | 388 (55.3%) | | 198 (62.3%) | 334 (59.1%) | |
| Not Answered | 13 (1.5%) | 5 (2.7%) | 8 (1.1%) | | 4 (1.3%) | 9 (1.6%) | |
| **Did you contact the study team by email?** | | | | 0.01* | | | 0.011* |
| Yes | 581 (65.8%) | 134 (73.6%) | 447 (63.8%) | | 190 (59.7%) | 391 (69.2%) | |
| No | 283 (32.0%) | 42 (23.1%) | 241 (34.4%) | | 122 (38.4%) | 161 (28.5%) | |
| Not Answered | 19 (2.2%) | 6 (3.3%) | 13 (1.9%) | | 6 (1.9%) | 13 (2.3%) | |
| **The study staff treated me with respect** | | | | 0.78† | | | <0.001† |
| All of the time | 695 (78.7%) | 141 (77.5%) | 554 (79.0%) | | 230 (72.3%) | 465 (82.3%) | |
| Most of the time | 22 (2.5%) | 5 (2.7%) | 17 (2.4%) | | 8 (2.5%) | 14 (2.5%) | |
| Some of the time | 3 (0.3%) | 0 (0%) | 3 (0.4%) | | 1 (0.3%) | 2 (0.4%) | |
| A little of the time | 1 (0.1%) | 0 (0%) | 1 (0.1%) | | 0 (0.0%) | 1 (0.2%) | |
| None of the time | 0 (0%) | 0 (0%) | 0 (0%) | | 0 (0%) | 0 (0%) | |
| Not Answered | 162 (18.3%) | 36 (19.8%) | 126 (18.0%) | | 79 (24.8%) | 83 (14.7%) | |
| **I felt the study staff was approachable** | | | | 0.54† | | | 0.001† |
| All of the time | 677 (76.7%) | 135 (74.2%) | 542 (77.3%) | | 228 (71.7%) | 449 (79.5%) | |
| Most of the time | 38 (4.3%) | 9 (4.9%) | 29 (4.1%) | | 10 (3.1%) | 28 (5.0%) | |
| Some of the time | 3 (0.3%) | 2 (1.1%) | 1 (0.1%) | | 1 (0.3%) | 2 (0.4%) | |
| A little of the time | 2 (0.2%) | 0 (0%) | 2 (0.3%) | | 1 (0.3%) | 1 (0.2%) | |
| None of the time | 0 (0%) | 0 (0%) | 0 (0%) | | 0 (0%) | 0 (0%) | |
| Not Answered | 163 (18.5%) | 36 (19.8%) | 127 (18.1%) | | 78 (24.5%) | 85 (15.0%) | |
| **I was satisfied with the answers I received from the study staff to my questions** | | | | 0.74† | | | 0.003† |
| All of the time | 639 (72.4%) | 128 (70.3%) | 511 (72.9%) | | 214 (67.3%) | 425 (75.2%) | |
| Most of the time | 65 (7.4%) | 15 (8.2%) | 50 (7.1%) | | 22 (6.9%) | 43 (7.6%) | |
| Some of the time | 14 (1.6%) | 3 (1.6%) | 11 (1.6%) | | 5 (1.6%) | 9 (1.6%) | |
| A little of the time | 1 (0.1%) | 0 (0%) | 1 (0.1%) | | 0 (0.0%) | 1 (0.2%) | |
| None of the time | 2 (0.2%) | 0 (0%) | 2 (0.3%) | | 0 (0.0%) | 2 (0.4%) | |

(*Continued*)

**Table 6.** (Continued)

| | Total (n = 883) | Age 30–50 (n = 182) | Age 70 and over (n = 701) | P-value | Male (n = 318) | Female or non-binary (n = 565) | P-value |
|---|---|---|---|---|---|---|---|
| Not Answered | 162 (18.3%) | 36 (19.8%) | 126 (18.0%) | | 77 (24.2%) | 85 (15.0%) | |

*Chi-squared test for trend (exact test used when expected cell counts were low)

†Chi-squared test for dichotomized response categories: "All of the time" or "Most of the time" vs. others (including "Not Answered")

which was a double negative or because the older cohort were more likely to contact the study team for instructions on how to do this.

The study procedures involved the need to use lancets to collect DBS. In our survey, 37% of the younger and 43% of the older cohort (40% men, 43% women) described difficulty with this procedure at least a little of the time. In order to improve this process, the team was available by email or phone to address concerns. There were many such contacts at the beginning of the study, which diminished with time. In response, the study team developed a short video to demonstrate how to use the lancets. This was not available at the time of the first sampling but in the survey 70% participants responded that they found the video helpful. Before it was available online, we sent an email with a picture of the lancet and its proper usage. We learned from this experience and ensured the instructional video for the rapid antigen test was readily available when the sub-study was initiated. There were fewer calls expressing difficulty with this procedure and the majority found the antigen tests easy to do and record.

In the survey 39% of both age and sex cohorts reported difficulty in getting sufficient blood to put on the DBS collection card. With the next shipment of DBS collection kits, the study team included two additional lancets of a different needle size. However, this created some confusion as their appearance was different and resulted in an increased number of contacts to the study team. Once the study participants were accustomed to the new lancets, we received feedback they worked better. It is important when changes are made to study procedures, no matter how trivial, that the reasons for the changes are explained, and appropriate instructions included.

Real time, non-judgmental, and supportive communication was essential to the success of this study. As anticipated, more of the older cohort preferred telephone and the younger cohort email as the method of choice to communicate with the study team. When designing future decentralized studies, investigators need to be prepared for the amount of communication that may be needed to help participants navigate through research procedures. We provided constant and consistent communication with responses typically immediate or within a few hours which led to high satisfaction. There were times when the study team had difficulty in responding to the burden of inquiries. Nonetheless, we feel that this was important to maintaining the engagement of the participants and budgeting of time and personnel needs to be considered for this type of trial. Email reminders throughout the study procedures via the study portal was also a positive feature reported by our participants in the survey.

As a decentralized clinical trial during the Covid-19 pandemic, the study website was being updated on an ongoing basis, with modifications made based on the availability and changing guidelines for additional vaccine doses, and the subsequent posting of participant antibody response results. In order for study participants to view these updates, they had to refresh and/or clear their cache each time. Some participants had difficulty with this and contacted the study team to walk them through the process.

The study was designed to be totally decentralized but this approach may be limiting for studies where physical exams or additional blood tests are required (although these challenges can be overcome with adequate resources) [29,30]. It does demonstrate how many study activities can be completed remotely and were acceptable to participants. As we did not have a comparative arm wherein all procedures and questionnaires were performed at a study site, and as many of our participants had never previously participated in research, we cannot conclude which method would be preferred. Given the need to participate through an electronic device, our study could not be generalizable to those who were unable to use such devices or who could not communicate in English and thus is not generalizable to the Ontario population receiving Covid-19 vaccines. The decentralized study did enable us to recruit a large number of participants in a short period time from around the province [31]. and enabled the participation of those who could not attend research facilities due to travel or mobility restrictions. We believe this to how aided in recruitment of a large number of elderly persons who might otherwise not be able to participate [32,33]. As we depended on self -report, there is no source documentation as is a feature of traditional clinical trials.

## Conclusion

STOPCoV was designed in the midst of the global Covid-19 pandemic. The study was virtually accessible for both younger and older participants and satisfaction with the study procedures and communication was high. The ease of remote access allowed study activities to be performed during Covid-19 restrictions and enabled participation of those living remotely from a study center. Despite digital challenges, study retention remains high. Availability of study staff to assist with procedures and to answer questions was essential. Instructional videos to guide self-sampling were valued. Research ethics boards are acknowledging that consent and study procedures can be done remotely.

## Acknowledgments

We acknowledge the additional members of the STOPCoV (Safety and Efficacy of Preventative COVID vaccines) Research team who have contributed to the main study:

Karen Colwill, PhD (Staff Scientist, Network Biology Collaborative Centre Manager, Lunenfeld-Tanenbaum Research Institute, Sinai Health System, University of Toronto, Toronto, Ontario, Canada)

Paula Rochon, MD (Women's College Research Institute, Women's College Hospital, University of Toronto, Toronto, Ontario, Canada)

Michael Brudno, PhD (Department of Computer Science, University of Toronto, Toronto, Ontario, Canada)

Janet Raboud, PhD (Emeritus Scientist, Toronto General Hospital Research Institute, University of Toronto, Toronto, Ontario, Canada)

Allison McGeer, MD (Mount Sinai Hospital, Department of Infection Control, University of Toronto, Toronto, Ontario, Canada)

Amit Oza, MD (University Health Network, Princess Margaret Hospital, Department of Medicine, University of Toronto, Toronto, Ontario, Canada)

Christopher Graham, MD (Trillium Health Partners, Department of Medicine, University of Toronto, Toronto, Ontario, Canada)

Amanda Silva, BSc (DATA Team, University Health Network, University of Toronto, Toronto, Ontario, Canada)

Jacqueline Simpson, BA (Healthcare Human Factors, University Health Network, University of Toronto, Toronto, Ontario, Canada)

Roaya Monica Dayam, PhD (Lunenfeld-Tanenbaum Research Institute, Sinai Health System, University of Toronto, Toronto, Ontario, Canada)

Adrian Pasculescu, PhD (Lunenfeld-Tanenbaum Research Institute, Sinai Health System, University of Toronto, Toronto, Ontario, Canada)

## Author Contributions

**Conceptualization:** Sharon Walmsley.

**Data curation:** Rizani Ravindran, Dorin Manase, Laura Parente.

**Formal analysis:** Leah Szadkowski, Leif Erik Lovblom.

**Funding acquisition:** Sharon Walmsley.

**Methodology:** Rizani Ravindran, Rosemarie Clarke, Qian Wen Huang, Sharon Walmsley.

**Project administration:** Rizani Ravindran, Rosemarie Clarke.

**Software:** Dorin Manase, Laura Parente.

**Supervision:** Sharon Walmsley.

**Visualization:** Qian Wen Huang.

**Writing – original draft:** Rizani Ravindran.

**Writing – review & editing:** Leah Szadkowski, Rosemarie Clarke, Qian Wen Huang, Dorin Manase, Laura Parente, Sharon Walmsley.

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
