## [Decision Letter · Decision Letter 0]

16 Jan 2023

PDIG-D-22-00252

Decentralized study of COVID Vaccine Antibody Response (STOPCoV): Results of a participant satisfaction survey.

PLOS Digital Health

Dear Dr. Walmsley,

Thank you for submitting your manuscript to PLOS Digital Health. After careful consideration, we feel that it has merit but does not fully meet PLOS Digital Health's publication criteria as it currently stands. Therefore, we invite you to submit a revised version of the manuscript that addresses the points raised during the review process.

Please submit your revised manuscript within 60 days Mar 17 2023 11:59PM. If you will need more time than this to complete your revisions, please reply to this message or contact the journal office at digitalhealth@plos.org. Please include the following items when submitting your revised manuscript:

We look forward to receiving your revised manuscript.

Kind regards,

Shlomo Berkovsky

Section Editor

PLOS Digital Health

Journal Requirements:

Additional Editor Comments (if provided):

Apologies for the delay - we struggled to find qualified reviewers for this submission.

The paper was eventually reviewed by 2 reviewers and both recommended Major Revision. I've gone through the paper and the reviews and agree with this recommendation. The key issues I'd recommend to address in the revision refer to the presentation of the method and statistical analysis.

Hope the reviews help you improve the submission.

Reviewers' comments:

Reviewer's Responses to Questions

**Comments to the Author**

1. Does this manuscript meet PLOS Digital Health’s publication criteria? Is the manuscript technically sound, and do the data support the conclusions? The manuscript must describe methodologically and ethically rigorous research with conclusions that are appropriately drawn based on the data presented.

Reviewer #1: Yes

Reviewer #2: Partly

2. Has the statistical analysis been performed appropriately and rigorously?

Reviewer #1: Yes

Reviewer #2: No

3. Have the authors made all data underlying the findings in their manuscript fully available (please refer to the Data Availability Statement at the start of the manuscript PDF file)?

Reviewer #1: Yes

Reviewer #2: No

4. Is the manuscript presented in an intelligible fashion and written in standard English?

Reviewer #1: Yes

Reviewer #2: Yes

5. Review Comments to the Author

Reviewer #1: The revised manuscript was attached in a word file.

Reviewer #2: In the manuscript entitled “Decentralized study of COVID Vaccine Antibody Response (STOPCoV): Results of a participant satisfaction survey.”, the Authors performed a sub-study from the main STOPCoV study to determine the satisfaction in decentralized study and compare the effect of age (70+ vs 30-50 years old).

The Authors major findings show a high overall satisfaction and moderate problems regarding procedure and blood collection.

This research addresses an important topic regarding the use of decentralized and web-based clinical trial, raised from SARS-CoV-2. However, there are some major points that need to be addressed:

Major Points:

- The study lacks of a formal analysis of Likert scale to compare age groups responses. Please provide statistical analysis and include the main results in Abstract, Results and discuss them in the Discussion sections.

- The dataset is highly unbalanced towards older population (21% 30-50 years old). This evidence should be discussed to improve participation of younger age in future studies. 

Minor Points:

- A part of the discussion should cover the pros and cons of decentralized studies in respect to on-site research studies.

6. PLOS authors have the option to publish the peer review history of their article (what does this mean?). If published, this will include your full peer review and any attached files.

**Do you want your identity to be public for this peer review?** For information about this choice, including consent withdrawal, please see our Privacy Policy.

Reviewer #1: No

Reviewer #2: No

---

## [Decision Letter · Decision Letter 1]

28 Feb 2023

PDIG-D-22-00252R1

Decentralized study of COVID Vaccine Antibody Response (STOPCoV): Results of a participant satisfaction survey.

PLOS Digital Health

Dear Dr. Walmsley,

Thank you for submitting your manuscript to PLOS Digital Health. After careful consideration, we feel that it has merit but does not fully meet PLOS Digital Health's publication criteria as it currently stands. Therefore, we invite you to submit a revised version of the manuscript that addresses the points raised during the review process.

Please submit your revised manuscript within 30 days Mar 30 2023 11:59PM. If you will need more time than this to complete your revisions, please reply to this message or contact the journal office at digitalhealth@plos.org. Please include the following items when submitting your revised manuscript:

We look forward to receiving your revised manuscript.

Kind regards,

Shlomo Berkovsky

Section Editor

PLOS Digital Health

Journal Requirements:

2. Please send a completed 'Competing Interests' statement, including any COIs declared by your co-authors. If you have no competing interests to declare, please state "The authors have declared that no competing interests exist". Otherwise please declare all competing interests beginning with the statement "I have read the journal's policy and the authors of this manuscript have the following competing interests:"

3. Please ensure that Funding Information and Financial Disclosure Statement are matched.

Additional Editor Comments (if provided):

Minor revisions/additions suggested by one of the reviewers. Please address these to get the paper to a publishable form.

Reviewers' comments:

Reviewer's Responses to Questions

**Comments to the Author**

1. If the authors have adequately addressed your comments raised in a previous round of review and you feel that this manuscript is now acceptable for publication, you may indicate that here to bypass the “Comments to the Author” section, enter your conflict of interest statement in the “Confidential to Editor” section, and submit your "Accept" recommendation.

Reviewer #1: All comments have been addressed

Reviewer #2: All comments have been addressed

2. Does this manuscript meet PLOS Digital Health’s publication criteria? Is the manuscript technically sound, and do the data support the conclusions? The manuscript must describe methodologically and ethically rigorous research with conclusions that are appropriately drawn based on the data presented.

Reviewer #1: Yes

Reviewer #2: Yes

3. Has the statistical analysis been performed appropriately and rigorously?

Reviewer #1: Yes

Reviewer #2: Yes

4. Have the authors made all data underlying the findings in their manuscript fully available (please refer to the Data Availability Statement at the start of the manuscript PDF file)?

Reviewer #1: Yes

Reviewer #2: Yes

5. Is the manuscript presented in an intelligible fashion and written in standard English?

Reviewer #1: Yes

Reviewer #2: Yes

6. Review Comments to the Author

Reviewer #1: Dear STOPCoV study authors,

Congratulations on the significant improvement of your manuscript. Here are some considerations regarding the methodology and results presented.

1- Present the demographic characteristics of the region studied, since PLOS Digital Health is a journal of international reach. I strongly believe that this information should be present in its methodology. 

2- Since this is a satisfaction study, we believe that the stratification of subjects (Male, Female, and non-binary) is fundamental to understanding the demographic delineation of the population. I understand that from a statistical point of view the ideal would be to group the information. However, in epidemiological studies the richness of detail must be maintained in the analysis. Therefore, I recommend stratifying the information. 

3- Add in Table 01 information regarding the self-declaration of race of the subjects included, since this is not described in the methodology or in the descriptive results.

Reviewer #2: I have carefully read the new version of the article "Decentralized study of COVID Vaccine Antibody Response (STOPCoV): Results of a participant satisfaction survey". The Authors made several improvements, especially in the statistical part. All my concerns have been properly answered.

7. PLOS authors have the option to publish the peer review history of their article (what does this mean?). If published, this will include your full peer review and any attached files.

**Do you want your identity to be public for this peer review?** For information about this choice, including consent withdrawal, please see our Privacy Policy. 

Reviewer #1: No

Reviewer #2: No

---

## [Editor Report · Decision Letter 2]

23 Mar 2023

Decentralized study of COVID Vaccine Antibody Response (STOPCoV): Results of a participant satisfaction survey.

PDIG-D-22-00252R2

Dear Dr. Walmsley,

We are pleased to inform you that your manuscript 'Decentralized study of COVID Vaccine Antibody Response (STOPCoV): Results of a participant satisfaction survey.' has been provisionally accepted for publication in PLOS Digital Health.

Best regards,

Shlomo Berkovsky

Section Editor

PLOS Digital Health